# TRPA1 as Target in Myocardial Infarction

**DOI:** 10.3390/ijms24032516

**Published:** 2023-01-28

**Authors:** Clara Hoebart, Attila Kiss, Patrick M. Pilz, Petra L. Szabo, Bruno K. Podesser, Michael J. M. Fischer, Stefan Heber

**Affiliations:** 1Center for Physiology and Pharmacology, Medical University of Vienna, 1090 Vienna, Austria; 2Center for Biomedical Research and Translational Surgery, Medical University of Vienna, 1090 Vienna, Austria

**Keywords:** cardiomyocytes, sensory neurons, ischemia, reperfusion-injury

## Abstract

Transient receptor potential cation channel subfamily A member 1 (TRPA1), an ion channel primarily expressed on sensory neurons, can be activated by substances occurring during myocardial infarction. Aims were to investigate whether activation, inhibition, or absence of TRPA1 affects infarcts and to explore underlying mechanisms. In the context of myocardial infarction, rats received a TRPA1 agonist, an antagonist, or vehicle at different time points, and infarct size was assessed. Wild type and TRPA1 knockout mice were also compared in this regard. In vitro, sensory neurons were co-cultured with cardiomyocytes and subjected to a model of ischemia-reperfusion. Although there was a difference between TRPA1 activation or inhibition in vivo, no experimental group was different to control animals in infarct size, which also applies to animals lacking TRPA1. In vitro, survival probability of cardiomyocytes challenged by ischemia-reperfusion increased from 32.8% in absence to 45.1% in presence of sensory neurons, which depends, at least partly, on TRPA1. This study raises doubts about whether TRPA1 is a promising target to reduce myocardial damage within a 24 h period. The results are incompatible with relevant enlargements of infarcts by TRPA1 activation or inhibition, which argues against adverse effects when TRPA1 is targeted for other indications.

## 1. Introduction

Cardiovascular diseases are the leading causes of death in Europe and the US [1,2], whereby myocardial infarction (MI) represents a major cause of mortality and morbidity and a huge socioeconomic burden. For MI treatment, early reperfusion of the occluded coronary artery is vital to limit myocardial damage [3,4]. However, reperfusion per se causes additional injury to the myocardium, which together with the damage caused by ischemia represents ischemia-reperfusion (IR) injury [4]. Understanding the mechanism of myocardial IR injury might help to improve the survival of cardiomyocytes and thereby improve patient outcomes.

Transient receptor potential cation channel subfamily A member 1 (TRPA1) is a non-selective cation channel that is mainly activated by electrophilic substances and therefore a sensor for potentially noxious molecules [5]. Such substances are released during ischemia and it has been described that TRPA1 acts as a sensor of hypoxia [6,7], as well as for reactive oxygen and nitrogen species [8,9,10,11]. TRPA1 is primarily expressed in nociceptive sensory neurons [10,12], but has also been found in other cell types, including in the cardiovascular system [9].

Upon activation, TRPA1 opens and allows cation influx, leading to depolarization and admission of calcium [8]. IR might activate TRPA1 on sensory neurons, and these might in turn release substances, affecting the densely innervated adjacent myocardium [13]. Thus, pharmacological TRPA1 activation could be beneficial in the context of IR. 

In addition, there was evidence that TRPA1 is expressed by cardiomyocytes [9]. Conceptually, this would increase Ca^2+^-overload during IR, having detrimental effects. From this perspective, it seemed also possible that pharmacological TRPA1 inhibition might be helpful in the context of IR. Notably, we demonstrated that there is no functional TRPA1 expression in cardiomyocytes in 2021 [14]. 

However, when this study was conceived in 2018, the available evidence regarding the role of TRPA1 in myocardial infarction was limited. There was one report indicating a reduction of infarct size in acute MI in rats for the treatment with TRPA1 agonists. This was supplemented by experiments in isolated cardiomyocytes that were challenged with hypoxia and subsequent reoxygenation, suggesting protective effects of TRPA1 agonists in vitro [15]. Notably, in this study, the agonists were given to rats exclusively prior to ischemia, limiting translatability to patients with acute myocardial infarction. 

Based on this prior knowledge, we aimed to investigate whether TRPA1 agonists or antagonists can confer beneficial effects in the context of myocardial IR-injury depending on the application time point, hypothesising that the effects would lead to opposing directions of increase and decrease of infarct size (primary outcome parameter) and be dependent on the time of intervention. For confirmatory purposes, we aimed to explore whether TRPA1^−/−^ mice show altered myocardial damage due to ischemia, assuming the effects to have the same direction as pharmacological inhibition. Based on the absence of TRPA1 in cardiomyocytes, we aimed to investigate mechanisms by which sensory neurons interact with cardiomyocytes in the context of IR in vitro. To this end, a cellular IR model was developed, in which the potential interactions between sensory neurons and cardiomyocytes were addressed. Regarding these in vitro experiments, we hypothesised that the presence of sensory neurons would increase the survival probability of cardiomyocytes in mimicked IR conditions and that this would be dependent on TRPA1 activation on sensory neurons. 

## 2. Results

### 2.1. Neither Pharmacological TRPA1 Activation Nor Inhibition Nor Absence of TRPA1 Protected the Heart against IR-Injury

Sprague–Dawley rats were subjected to 30 min of ischemia by ligation of the left coronary artery followed by 24 h of reperfusion. The rats were either treated systemically (i.v. bolus injection) with the TRPA1 agonist JT010 60 min (Pre) or 5 min prior to ischemia (Early) or 5 min prior to the onset of reperfusion (Late). Additional groups were treated with the TRPA1 antagonist A-967079 at the Early or Late time point (Figure 1A). Some rats had to be excluded from the analysis, but the proportion of rats that were excluded did not differ significantly between groups (*p* = 0.44, Table A1). The area at risk (AAR) in percent of the left ventricle was similar among the groups, with an overall mean of 32% (Figure 1B and Figure A1A). Based on the pre-specified analysis, there was no evidence for any of the treatments to change the infarct size compared to the 47.9% infarct size of the area at risk in the control group (effect of group *p* = 0.071). Exploratory analyses showed that compared to control, all groups exposed to JT010 combined were not different (all JT010 vs. control: 3.3%, CI: −2.5–9.2%, *p* = 0.26), and this applies also to all groups exposed to A-967079 (all A-967079 vs. control: −3.8%, CI: −10.3–2.6%, *p* = 0.24). Notably, infarct size after TRPA1 agonist treatment (4.9%, 4.4% and 0.7% above control), was higher than after TRPA1 antagonist treatment (3.1% and 4.6% below control, *p* = 0.005, Figure 1C).

In C57BL/6J wild type and TRPA1^−/−^ mice, the left coronary artery was permanently ligated to induce a myocardial infarction and hearts were analysed 24 h later. The area at risk in percent of the left ventricle was similar for both genotypes, with a mean of approximately 29% (Figure 2A and Figure A1B), and the number of mice that had to be excluded from the analysis was not different between the two groups (*p* = 0.42, Table A2). The applied statistical model resulted in an estimated mean infarct size of 69.1% in wild type, and 59.0% in TRPA1^−/−^ mice (*p* = 0.31, Figure 2B). These results do not support a prominent role of TRPA1 in myocardial damage due to myocardial infarction. 

### 2.2. Sensory Neurons Increased the Survival of Cardiomyocytes in a Cellular Model of Ischemia-Reperfusion

Cardiomyocytes with or without sensory neurons from dorsal root ganglia (DRG) were cultured for the cellular model of IR, including 2 h of mimicked ischemia and 0.5 h mimicked reperfusion. These periods were experimentally determined to result in a cardiomyocyte (CM) survival of ~50% in the absence of sensory neurons (Figure 3A and Figure A2). Cardiomyocyte survival was assessed on a morphological basis by taking transmission widefield images before and after mimicked IR (Figure A3A,B). Sensory neurons showed a nearly identical survival rate of more than 85% in control and IR conditions (*p* = 0.59), with a seeding density of approximately 128 cells/mm^2^ (Figure A5). 

In a 2.5 h control in cell culture, 77.4% of cardiomyocytes were still present in healthy morphology, in the chosen model for IR the survival of cardiomyocytes decreased to 32.8% (*p* < 0.001, Figure A4A). The effect of ischemia on the survival probability was dependent on the presence of sensory neurons (sensory neuron*ischemia interaction *p* = 0.005). In co-culture with sensory neurons, the survival probability in IR conditions was increased from 32.8% to 45.1% (*p* = 0.015), corresponding to a factorial change of 1.38-fold. 

To test whether TRPA1 is involved in the interaction of the two cell types, sensory neurons derived from TRPA1^−/−^ animals were usedReceptor specificity was tested by probing sensory neurons from wild type animals in parallel. Regarding TRPA1^−/−^ sensory neurons, the effect of IR on the survival probability of CM tended to differ between the three conditions with no sensory neurons, wild type sensory neurons and TRPA1^−/−^ sensory neurons (ischemia*sensory neuron interaction *p* = 0.07). Additionally, in this smaller experimental sample performed simultaneously, cardiomyocyte survival under IR of 23.9% was increased to 39.0% in the presence of wild type sensory neurons. In contrast, there was no significant increase in CM survival when co-cultured with TRPA1^−/−^ sensory neurons (30.6%, *p* = 0.31). While the nominal difference of cardiomyocyte survival of 8.4% between wild type (WT) and TRPA1^−/−^ sensory neurons did not reach statistical significance, we would consider such an effect size relevant (*p* = 0.17, Figure 4A,B). 

In experiments conducted to test TRPV1^−/−^ sensory neurons regarding their ability to protect cardiomyocytes from ischemic damage, the effect of IR on the survival probability of CM differed between the three groups (no sensory neurons, sensory neurons of wild type and TRPV1^−/−^ animals, ischemia*sensory neuron interaction *p* = 0.029). However, contrasts showed that the survival probability of cardiomyocytes was similar in the co-culture of cardiomyocytes with TRPV1^−/−^ sensory neurons to co-culture with WT neurons (*p* = 0.70). In control conditions, there was no difference between the presence of wild type and TRPV1^−/−^ sensory neurons on the survival probability of cardiomyocytes (*p* = 0.44, Figure 4C,D). Results of different wells within each experimental day are shown in Figure A4. Overall, these experiments showed that sensory neurons could increase the survival of cardiomyocytes in IR. This appeared, at least partially, to be dependent on the presence of TRPA1.

## 3. Discussion

In the context of a myocardial infarction model, neither TRPA1 activation or inhibition at different time points in rats, nor the absence of TRPA1 in mice conferred relevant protective effects in this study. In vitro, sensory neurons cultured together with cardiomyocytes can exert a protective effect on the latter.

The study evaluated the effects of treatment with a TRPA1 agonist and antagonist in experimental acute myocardial infarction. At the time the study was planned, the literature suggested that in the heart TRPA1 is expressed at least on neurons and cardiomyocytes. Both inhibition and activation of TRPA1 were considered to potentially limit myocardial damage. Inhibition might have been helpful as it could limit calcium overload of cardiomyocytes during myocardial reperfusion caused by TRPA1 being activated by reactive oxygen species and other substances released during IR. Time points ‘Early’ and ‘Late’ were performed to be able to differentiate between TRPA1 inhibition during reperfusion (‘Early’ and ‘Late’) compared to a potential additional effect when the antagonist is also present during ischemia (only ‘Early’). A time point ‘Pre’ was omitted, as no baseline TRPA1 activation was assumed, and therefore a ‘Pre’-application of the antagonist would have yielded similar results to ‘Early’. With respect to pharmacological activation, channel desensitisation of myocardial TRPA1 might have been helpful, hence the application of JT010 well before the myocardial infarction (time point ‘Pre’). Additionally, neuronal activation might have released protective messengers, e.g. neuropeptides (time points ‘Early’ & ‘Late’). The only available in vivo study at the time this study was planned showed that the TRPA1 activators ASP 7663 and optovin applied before ischemia reduced infarct size in a rat model of acute myocardial infarction with 30 min ischemia followed by 2 h reperfusion [15]. In principle, this study is similar to the time point ‘Early’ in the present study, yet only 2 h of reperfusion, during which the myocardial damage may not have reached its final size, is a limitation. We tried to overcome this limitation with a study design including 24 h of reperfusion. Further, the prior study was extended by multiple application time points, differentiating between effects at ischemia and reperfusion or only the latter.

From the current perspective, all hypotheses implying an effect of TRPA1 in cardiomyocytes became questionable, given that we found no functional TRPA1 on cardiomyocytes by multiple approaches [14]. Nevertheless, while we were conducting the present study, further reports were published implying such a direct effect of TRPA1 on cardiomyocytes in ischemia [16,17]. As such, the results of these studies are to be interpreted with caution, and the mechanistic basis of these observations remains unclear. One of these studies describes a reduction of infarct size in TRPA1^−/−^ mice after 30 min of ischemia followed by 24 h of reperfusion [16]. Their IR model resulted in infarct sizes of 34% (WT) vs. 14% (TRPA1^−/−^). Our experiments investigated permanent ligation of the left coronary artery, which resulted in infarct sizes of 69% (WT) and 59% (TRPA1^−/−^). In both studies animals lacking TRPA1 have a lower infarct size, and their result of 20% difference in infarct size is well within the confidence interval of our result. This suggests that TRPA1^−/−^ mice may be modestly resistant to ischemic damage. However, the responsible TRPA1-expressing cell type remains elusive and further studies should address the role of TRPA1 in post MI adverse remodelling. Another study exclusively contains in vitro experiments with isolated cardiomyocytes and comes to the conclusion that activation of cardiomyocyte TRPA1 promotes survival in ischemia [17]. This is implausible given the absence of functional channels, but also our in vivo results argue against protective effects of TRPA1 activation during ischemia. Based on the width and coverage of the 95% confidence intervals, the present study is compatible with a minor increase of infarct size due to JT010, yet hardly with infarct size reductions. In the case of A-967079, the respective upper limit is only 4.7% when applied ‘Early’. Additionally, the data are compatible with an infarct size reduction of up to 11.8% due to TRPA1 inhibition (lower limit of 95% CI Control vs. A-967079 ‘Late’). In the synopsis of published data and the results presented here, it appears that exogenous TRPA1 activation is most likely not helpful and potentially detrimental. Pharmacological TRPA1 inhibition might at best have small infarct-limiting effects of unclear relevance for the human situation. It should be noted that the results are not compatible with a relevant adverse effect of TRPA1 inhibitors in case of an ischemic cardiac event, which is helpful considering their pharmacological use for other indications. 

Furthermore, based on a recent study [18], TRPA1 inhibitors might have beneficial effects when given long-term after myocardial infarction by improving myocardial function, reducing fibrosis, and promoting angiogenesis. However, another study described that both TRPA1 knockout and treatment with a TRPA1 agonist led to a reduction of survival of mice 3 to 7 days post MI compared to wild type non-treated mice [19]. These outcomes are surprising, and it seems unlikely that both activation and knockout of TRPA1 decreases mouse survival post-MI. These studies focused on the chronic phase after MI, and are therefore not opposed to the present study, which only addressed the role of TRPA1 in acute MI. It might also be worthwhile to investigate the involvement of sensory neurons and specifically TRPA1 in ischemic preconditioning, as different neuronal and humoral components are involved, and its effects might be enhanced [20]. 

Somewhat contradictory results regarding ischemia of other organs, such as the kidneys, eyes, and brain indicate open questions. In the kidneys [21], there are also inconsistent results about whether TRPA1 activation exacerbates or improves acute kidney injury [22,23]. For IR damage of the retina, both knockout of TRPA1 or treatment with TRPA1 antagonists had protective effects as measured by retinal cell numbers and thickness [24,25]. In an ischemic stroke model knockout of TRPA1 in cerebral artery endothelial cells led to an increase in cerebral infarcts, whereas treatment with TRPA1 agonists reduced infarct sizes [26]. In rat optic nerves, TRPA1 antagonists reduced myelin damage after ischemia [27]. In a model for white matter ischemia, the damage of oligodendrocytes was reduced by TRPA1 antagonists [28]. Overall, effects seem to be cell type dependent and variable between models. 

To uncover mechanisms underlying a possible interplay between sensory neurons and cardiomyocytes under IR conditions, we developed a co-culture of primary adult murine sensory neurons with cardiomyocytes. Earlier studies only showed co-culture of DRG explants and cardiomyocytes of neonatal rats [29]. This newly developed co-culture was then exposed to an IR model, which mimics the pathophysiology of MI in several aspects. This includes anoxia, a reduction of glucose levels, a change in metabolite levels, a decreased pH during ischemia, and a reappearance of oxygen and glucose during reperfusion. Similar approaches were performed in cardiac cell lines, yet not in primary cardiomyocytes [30,31]. Other approaches used hypoxia [32,33,34], or cobalt chloride to mimic hypoxia [35,36] or changes in metabolism [37] to mimic ischemia, which does not cover the whole pathophysiology, including the reperfusion injury. We optimised the duration of simulated ischemia so that cardiomyocyte survival was approximately half of that under control conditions, which allows us to detect both an increase and a decrease in cell survival. The ischemia period for 50% survival of primary cardiomyocytes is shorter than those of cardiac cell lines [30,31]. The culture conditions are well-controlled by coating with a biologically well-defined peptide hydrogel which has mechanical properties within a narrow range and shows no batch-to-batch variation. This is in contrast to the commonly used Matrigel, which is derived from a mouse sarcoma cell line and contains unknown and potentially variable factors that could influence the culture. 

In line with the hypothesis that sensory neurons sense ischemia and in turn release cardioprotective substances, we found that the addition of sensory neurons increased the survival probability of cardiomyocytes exposed to the IR-model. This effect could be mediated by direct interaction of the two cell types or by soluble substances released by the sensory neurons. However, there was a tendency of sensory neurons to decrease the survival of cardiomyocytes in control conditions. The mechanism is unclear; there might be, for example, continuous substance release by sensory neurons or an effect of surface molecules which only occur on the cell bodies but not the axons of sensory neurons. The finding that sensory neurons were not harmed by this model of IR demonstrates that they exert their effect in a viable state.

The TRPA1 channel is activated by hypoxia and substances released during ischemia [7,11]. Therefore, we hypothesised that TRPA1 is involved in the increase of IR tolerance of cardiomyocytes in the presence of sensory neurons. However, while the point estimate of the protective effect of TRPA1-deficient sensory neurons was less than half that of wild type sensory neurons, the difference between these groups was not significant. Thus, the data do not provide robust evidence that the protective effects of sensory neurons on cardiomyocytes are dependent on sensory neuron TRPA1, but rather asks for independent confirmation by others. The findings are therefore also compatible with contributions of other possible mechanisms in ischemia pathophysiology [38]. 

Regarding the in vitro results of TRPV1, the point estimate of the protective effect of wild type sensory neurons and TRPV1 deficient ones were nearly identical. This is in contrast to a study showing that intrathecal shRNA-based silencing or pharmacological inhibition of TRPV1 leads to a reduction in myocardial infarct size in rats undergoing 30 min ischemia followed by 2 h reperfusion [39]. However, protective effects of TRPV1 activation in IR injury have been reported as well [40,41]. 

### 3.1. Limitations

The interactions of sensory neurons with cardiomyocytes require further investigation. The in vitro results suggest that sensory neurons can exert protective effects on cardiomyocytes in IR. Considering TRPA1-dependence, this might seem at odds with the results in vivo, where the role of TRPA1 seems marginal. However, the difference between these experimental approaches needs consideration. In the co-culture model, there were on average two neurons per cardiomyocyte, which does not reflect the ratio in vivo. Further, in vivo the infarct size was measured 24 h after MI induction, whereas in vitro the CM survival was determined directly after 2 h ischemia plus 30 min reperfusion. Finally, in a myocardial infarction in vivo there are other cell types present, including immune cells, fibroblasts, and sympathetic efferent axons, which are not recapitulated by the in vitro model. The observed infarct size, in the absence of any pharmacological intervention, was lower than anticipated when the study was planned, the pooled variances matched the expectation. Possible explanations might be the chosen anaesthetic protocol as well as the rat strain. The presented results regarding infarct size do not allow conclusions concerning a role of TRPA1 in arrhythmia, as indicated by prior studies [42,43,44]. 

### 3.2. Conclusions 

TRPA1 does not seem to play a major role in the cell damage inflicted by acute myocardial infarction. This is beneficial for the pharmacological use of TRPA1 antagonists, as a risk of cardiovascular side effects in the case of myocardial infarction appears unlikely. In vitro it was seen that sensory neurons increase the survival probability of CM in IR conditions, to which TRPA1 might contribute. It still needs to be clarified whether, and if so which, soluble factors might mediate a sensory neuron-cardiomyocyte interaction during IR, and whether this has potential to be translated to reduced IR injury in vivo. 

## 4. Materials and Methods

### 4.1. Animals

Adult male Sprague–Dawley rats and adult male C57BL/6J mice (both from the Department for Laboratory Animal Science and Genetics, Himberg, Austria), TRPA1^−/−^ and TRPV1^−/−^ mice were used. Heterozygous TRPV1^+/−^ mice were a generous gift from John B. Davis [45], and TRPA1^+/−^ animals from David P. Corey [46]. Animals were back-crossed on C57BL/6J for more than ten generations. Breeding, euthanasia, and all procedures of animal handling were performed according to regulations of animal care and welfare. Experiments were carried out in accordance with the European Communities Council Directive of 24 November 1986 (86/609/EEC). For in vitro experiments mice were euthanized by cervical dislocation, preceded by anaesthesia by exposure to isoflurane (CP Pharma, Burgdorf, Germany). Animals were housed with a 12:12 h light-dark cycle and access to food and water ad libitum.

### 4.2. In Vivo Experiments

For in vivo experiments, male adult (10–14 weeks) Sprague–Dawley rats, and male adult (10–14 weeks) C57BL/6J and TRPA1^−/−^ mice were used. Experiments were performed at the Centre for Biomedical Research and Translational Surgery, Medical University of Vienna. The experimental protocol was approved by the Austrian Ministry of Science, Research and Economy (BMWFW-66.009/0413-V/3b/2018), and conforms with the Guide for the Care and Use of Laboratory Animals, published by the US National Institutes of Health (NIH Publication No. 85–23, revised 1996).

### 4.3. Rat Model of Myocardial Ischemia-Reperfusion Injury

Briefly, rats were anaesthetised by intraperitoneal injection of a mixture of xylazine (4 mg/kg body weight, Bayer, Germany) and ketamine (100 mg/kg, Graeub, Bern, Switzerland), intubated (14-gauge tube), and ventilated (9 mL/kg, 75–85 strokes/min). Body temperature was continuously monitored using a rectal temperature probe and maintained at 37.5–38.5 °C using a thermostatic heating pad. The heart was exposed via a left thoracotomy and a ligature was placed around the left coronary artery for 30 min at a distance of 2–3 mm from the origin. Myocardial ischemia was associated with pallor of the myocardial area at risk and ST-elevation on ECG signal. Reperfusion was initiated following the 30 min of left coronary artery occlusion by removal of the snare and lasted for 24 h. The thoracotomy was closed with 4/0 monofilament sutures. Analgesia was initiated by intraperitoneal injection of piritramide (0.1 mL/kg, Hameln Pharma, Hameln, Germany) preoperatively, and piritramide was applied in drinking water as a postoperative analgesic regimen (30 mg piritramide with 30 mL of glucose 5% in 250 mL water).

### 4.4. TRPA1 Agonist and Antagonist Treatment of Rats

The TRPA1 agonist JT010 is the most potent commercially available TRPA1 agonist [47] and its specificity has been shown against other TRP channels known to be activated by electrophilic molecules. A-967079 is a potent antagonist of TRPA1, which is “>1000-fold selective over other TRP channels, and is >150-fold selective over 75 other ion channels, enzymes, and G-protein-coupled receptors” [48]. JT010 at 0.4 mg/kg body weight or A-967079 at 6.2 mg/kg body weight (both obtained from Sigma-Aldrich, St. Louis, MO, USA) were applied intravenously (left femoral vein). The agonist and antagonist were solved in DMSO (dimethyl sulfoxide, Sigma-Aldrich). These solutions were diluted to 68% DMSO content with PB-SIF (phosphate buffered-synthetic interstitial fluid) containing (in mM) 107.8 NaCl, 5 glucose, 3.5 KCl, 0.7 MgSO_4_ × 7H_2_O, 23 Na_2_HPO_4_, 9.6 Na-Gluconate, 7.6 saccharose, and 1.5 CaCl_2_ × 2H_2_O in distilled water. Treatment with JT010 was performed at the following time points: 60 (‘Pre’) or 5 min (‘Early’) prior to left coronary artery occlusion and 5 min prior to onset of the reperfusion (‘Late’). Treatment with A-967079 was performed at the ‘Early’ and ‘Late’ time points, as displayed in Figure 1A. A PB-SIF injection was performed at every time point without substance application to maintain the blinding of the experimenter. Application time points were randomised over experimental days including the control solution.

### 4.5. Mouse Model of Myocardial Infarction

The murine model of myocardial infarction has been described previously [49]. Briefly, male C57BL/6J and TRPA1^−/−^ mice were anaesthetised with a MMFK-mix (medetomidine 0.3 mg/kg (Orion, Espoo, Finland), midazolam 1 mg/kg (Accord Healthcare, Utrecht, Netherlands), fentanyl 0.03 mg/kg (Hameln Pharma), and ketamine 10 mg/kg (Livisto, Senden, Germany); intraperitoneal injection at 10 µL/g bodyweight). Heart and respiratory rate and temperature were monitored throughout the procedure and the pedal reflex was assessed. The chest was shaved, washed with antiseptic solution and a thoracotomy was performed in the fourth left intercostal space. The left ventricle was visualised, the pericardial sac removed, and the left coronary artery permanently ligated using a 7/0 monofilament suture (Peters surgical, Boulogne-Billancourt, France) at the site of its emergence from under the left atrium. Altered ECG and significant colour changes at the ischemic area were considered indicative of successful MI. The thoracotomy was closed with 6/0 monofilament sutures. After surgery the mice were injected with atipamezole (0.2 mL/kg, Orion) and naloxone (1 mL/kg, Amomed Pharma, Vienna, Austria) to antagonise the MMFK anaesthesia. The endotracheal tube was removed once spontaneous respiration resumed, and mice were placed on a warm pad maintained at 37 °C until they were completely awake and alert. Piritramide (Hameln Pharma, 15 mg of piritramide diluted in 20 mL 5% glucose) in the drinking water was used as postoperative analgesia.

### 4.6. Assessment of Myocardial Infarct Size

Myocardial infarct size was measured as described previously [50]. Briefly, 24 h after the initiation of reperfusion in rats and 24 h after initiation of permanent ischemia in mice, the animals were anaesthetised and ventilated as described above (see in the description of the infarct models) and the coronary artery was reoccluded in rats and 1.0–1.5 mL of 2% Evans Blue (Sigma-Aldrich) was injected in the left ventricle via the caudal vena cava to mark the ischemic myocardium (area at risk). The animals were euthanized by exsanguination and the heart was rapidly excised. The atria and the right ventricle were removed. The left ventricles were frozen for 20 min at −20 °C and cut into 5–7 slices perpendicular to the base-apex axis. These slices were scanned from both sides for the determination of the area at risk, weighed, and put in 1% triphenyltetrazolium chloride (Sigma-Aldrich) for 15 min at 37 °C to stain the viable myocardium as described previously [50]. After 24 h of incubation in 4% formaldehyde (Sigma-Aldrich), slices were scanned again from both sides, and the infarct size in percentage of the area at risk was determined by planimetry of the images (Photoshop 6.0; Adobe Systems, San Jose, CA, USA).

### 4.7. In Vitro Experiments—Chemicals and Solutions

The Ca^2+^ free solution used to isolate cardiomyocytes contains (in mM): 134 NaCl, 11 glucose, 4 KCl, 1.2 MgSO_4_, 1.2 Na_2_HPO_4_, 10 HEPES. This solution was buffered to pH 7.35 with NaOH and the myosin inhibitor 2,3-butanedione monoxime 10 mM was added. The ‘ischemic solution’ contained (in mM): 137 NaCl, 8 KCl, 0.49 MgCl_2_ × 6H_2_O, 0.9 CaCl_2_ × 2H_2_O, 4 HEPES, and 20 sodium lactate and is set to a pH of 6.2. The ‘reperfusion solution’ contained (in mM): 137 NaCl, 2 KCl, 1.5 MgCl_2_ × 6H_2_O, 0.9 CaCl_2_ × 2H_2_O, 4 HEPES, and 12 glucose, and is set to a pH of 8.8. NaCl and HEPES were obtained from Carl Roth (Karlsruhe, Germany), KCl, MgSO_4_, Na_2_HPO_4_, MgCl_2_ × 6H_2_O and sodium lactate were obtained from Sigma-Aldrich, CaCl_2_ × 2H_2_O was obtained from Merck, and glucose and NaOH from Thermo Fisher Scientific (Waltham, MA, USA).

### 4.8. Isolation of Primary Cardiomyocytes and Dorsal Root Ganglion Sensory Neurons

For cultures of sensory neurons, DRG from all spinal levels were excised and transferred to DMEM (Dulbecco’s Modified Eagle’s Medium, D5648, Sigma-Aldrich) containing 1% streptomycin/penicillin (Lonza, Basel, Switzerland) and 1% L-glutamine (Lonza), treated with 1 mg/mL collagenase (Sigma-Aldrich) and 3 mg/mL Dispase II (Roche, Mannheim, Germany) for 60 min at 37 °C. Enzymatically digested DRG were mechanically dissociated with a Pasteur pipette, centrifuged at 235 g for 5 min, and plated in 24 well plates coated with the Peptigel alpha 2-IKVAV, which is functionalized with laminin, diluted at 1:10 in Peptisol (Manchester Biogel, Manchester, UK) and then 1:1 in DMEM. Additional plates were coated for single cell culture of cardiomyocytes. DRG neurons were cultured in DMEM supplemented with 100 mg/mL streptomycin/penicillin (Lonza), 1% L-glutamine (Lonza), and 100 ng/mL mouse nerve growth factor (Alomone Labs, Tel Aviv, Israel). Neurons were cultured at 37 °C and 5% CO_2_ for 3 days with the daily exchange of medium.

Three days after the seeding of sensory neurons, primary cardiomyocytes were isolated. The isolation procedure was started by injecting a volume of 5 mL of ice-cold calcium-free solution into the ventricles to clean them from blood and stop the contractions. The heart was then surgically removed, suspended in the Langendorff apparatus by the aorta [51], retrogradely perfused at 37°C with a calcium-free solution for 3 min, and then with a calcium-free solution containing 0.17 mg/mL Liberase (Roche) for 18 min at 180 mL/min. The atria were discarded, and the ventricles were mechanically triturated with fine forceps. The cellular digestion solution was then incubated on a shaker at 37°C and was progressively exposed to increasing concentrations of calcium, reaching a maximum of 0.2 mM of calcium over one hour in five steps. A final step of mechanical trituration with a plastic Pasteur pipette was performed, isolated cardiomyocytes were centrifuged at 9 g for 3 min and resuspended with a glass pipette in CM medium containing M199 (M5017) supplemented with 10 µg/mL insulin, 5.5 g/mL transferrin and 5 ng/mL selenium, 0.1% bovine serum albumin, 10 mmol/l 2,3-butanedione monoxime, 1% penicillin/streptomycin, 1% L-glutamine (all Sigma-Aldrich), and 1X CD lipid (Thermo Fischer Scientific) adapted from Ackers-Johnson et al. [52].

For single cell culture, cardiomyocytes were seeded in the previously coated 24-well plates. For the co-culture, the isolated cardiomyocytes were seeded onto the sensory neuron cultures. After incubation at 37 °C and 5% CO_2_ for 30 min to ensure cardiomyocyte recovery and attachment to Peptigel, brightfield images were taken with an Olympus IX73-inverted microscope (Olympus, Tokyo, Japan) using a 4x objective.

### 4.9. Cellular Ischemia-Reperfusion Model 

To mimic ischemia, the cell cultures were temporarily deprived of oxygen and glucose (protocol visualised in Figure 3A). This was achieved with an “ischemic solution” containing no glucose but lactic acid, a pH adjusted to 6.2, and incubation in an anoxic environment in a Modular Incubator Chamber (Embrient, San Diego, CA, USA) flushed with gas containing 95% N_2_, 5% CO_2_ (Linde Gas, Dublin, Ireland) and maintained at 37 °C for 2 h. This time was determined in preliminary experiments and led to the survival of approximately 45% of CM, which is half the CM survival of control conditions (Figure A2). The plates were taken out of the Modular Incubator Chamber (Embrient) and “reperfusion solution” was added on top. With the addition of the reperfusion solution, the pH and glucose levels are balanced back to physiologic levels. The plates were then placed in the incubator at 37 °C and 5% CO_2_ for 30 min, thereafter a second brightfield image was acquired. For control conditions, the same steps were performed, but cells were only treated with CM medium and were placed in an incubator at 37 °C and 5% CO_2_ instead of the Modular Incubator Chamber.

### 4.10. Analysis of Cardiomyocyte Survival

The images taken before and after IR were analysed regarding the survival of CM using ImageJ [53]. Thereby cardiomyocyte survival was manually selected by morphology, with rounded up cells being counted as dead and elongated and rod-shaped CM being counted as alive, according to Mishra et al. [54]. Determination of cell viability was performed by the first author to keep the morphological criteria of dead vs. alive cells as similar as possible across experiments. Additionally, some cells disappeared from the field of view and were counted as lost. The level of damage in the lost cells cannot be judged, therefore the respective cells were not included in the analysis. The fraction of lost cells was similar in control and IR. Exemplary images are shown in Figure A3.

### 4.11. Analysis of Sensory Neuron Survival and Density 

Sensory neurons were isolated from WT mice and cultured for 3 days. Then, the medium was changed to CM medium, and the cells either underwent the cellular IR model or stayed in control conditions with CM medium, as described above. To detect cell death of sensory neurons, the cells were stained with the Annexin V-FITC apoptosis detection kit with 7-AAD (Tonbo biosciences, San Diego, CA, USA) with a protocol adapted from the manufacturer’s instructions. First, wells were washed with stain buffer two times, and then 250 µL Annexin V binding buffer, 12.5 µL FITC Annexin V conjugate, and 12.5 µL 7-AAD solution were added per well, and the plates were incubated at room temperature for 15 min in the dark. Next, 400 µL Annexin V binding buffer were added and the plates were imaged at 488 nm, 550 nm, and brightfield with an Olympus IX73-inverted microscope (Olympus) using a 4x objective. Image contrasts were set with unstained wells and for the analysis of sensory neuron survival, cells that were positive for Annexin V as an early damage marker, or 7-AAD as a late damage marker, or both were counted as dead. Additionally, the density of cells was calculated in cells/mm^2^.

### 4.12. Statistical Analyses and Sample Size Calculations

Statistical analysis was performed using IBM SPSS statistics 28 (Armonk, NY, USA). Before the analysis, rats were excluded in case they did not survive the surgery or the 24 h after, or in case the AAR was smaller than 15% or technical problems with the staining of the hearts occurred (Table A1). These criteria were pre-defined and the proportion of rats that were excluded was compared between groups by a Chi-square test.

For the analysis of the infarct size of rats treated with an antagonist or an agonist during myocardial infarction, a linear mixed model with the dependent variable “infarct size”, the factor “treatment groups”, the covariates “area at risk”, “body weight at day of operation”, and “body weight at day of harvest” was used. To account for the dependency of data generated at a certain experimental day, each day was used as a level of a random factor. The main effect of “treatment groups” was used to test the main hypothesis for which comparisons of each group with the control group were pre-specified. The contrasts comparing (i) all groups that received A-967079 with those that received JT010, (ii) all groups that received A-967079 with the control group and (iii) all groups that received JT010 with the control group were not pre-specified. Approximate normal distribution of residuals was inspected visually. The sample size was based on the following considerations: Based on previous experiments [50], we expected a mean ± SD infarct size of 65 ± 12% in untreated rats. We considered a 15% reduction of infarct size as relevant improvement, corresponding to 50% infarct size in a treated group. Considering the 5 pairwise comparisons with the control group, α was set from 0.05 to 0.01. Assuming the previously observed standard deviation, 17 animals/group were necessary to detect the relevant mean difference of 15% with a power of 80%, accepting the probability of a type I error of 5% after correction for multiplicity. Estimation of necessary sample size was performed with nQuery (Statstools, Los Angeles, CA, USA). We expected an attrition rate due to arrhythmia, impaired function, or surgical issues of 25%. Thus, 23 animals needed to be included for each of the 6 groups. Animals were randomized in blocks of 6 to the 6 treatment groups. On each experimental day, 6 animals were treated directly after each other. To avoid confounding by time of day or the order, sequences were generated in a way that each treatment occurred the same number of times at each period (Williams latin squares design). 

Before the analysis of data obtained from WT and TRPA1^−/−^ mice, animals were excluded in case they did not survive the surgery, had almost no visible infarct, probably due to insufficient artery ligation, or technical problems with the staining of the hearts as shown in Table A2. The proportion of mice that were excluded was compared between groups by Fisher’s exact test. For the analysis of the infarct size of WT and TRPA1^−/−^ mice, a linear model with the dependent variable “infarct size”, the factor “genotype”, the covariates “area at risk”, “body weight at day of operation”, and “body weight at harvest” was performed. The hypothesis was tested by the effect of “genotype”. Sample size was determined based on an expected standard deviation of infarct size of 10% in wild type mice, based on earlier experiments. We considered an infarct size change of 15% due to knockout of TRPA1 relevant. To detect this difference with a power of 80%, accepting the probability of a type I error of 5%, 9 mice per group were necessary. As above, sample size calculation was performed with nQuery.

For the analysis of the survival of cardiomyocytes in the cellular model of IR generalised linear mixed models with a binomial target distribution and a logit link corresponding to binary logistic regression was used. In all models, the unit of observation was a cardiomyocyte, and whether it survived or not was the binary target variable. On each experimental day, cardiomyocytes were isolated from a single mouse and the isolated cells were distributed over several wells. Thus, observations from one experimental day were related to each other, hence “experimental day” was used as random factor in all models. From each well on each experimental day, several images were taken from which cells were counted. Thus, “well” with number of the well as level and “photo” with number of the image as level were used as random factors in all models.

The fixed factors depended on the experiment. The model corresponding to the experiment displayed in Figure 3B,C and Figure A4A contained the two binary fixed factors “sensory neurons” with the levels “sensory neurons present in culture” or “sensory neurons not present in culture” and “ischemia” with the levels “ischemia” and “control”. The two factors were used in a factorial manner, i.e. with the interaction. To break down the significant interaction term, contrasts were used to estimate the effects of “sensory neuron” on survival within each level of “ischemia” and vice versa.

The model corresponding to the experiment displayed in Figure 4A,B and Figure A4B contained the binary factor “ischemia” and the factor “sensory neurons” with the three levels “no sensory neurons”, “wild type sensory neurons”, and “TRPA1 deficient sensory neurons”. The significant interaction between these two factors was broken down by appropriate contrasts. 

An analogous model was used to analyse data displayed in Figure 3C,D and Figure A4C, except TRPV1 instead of TRPA1 deficient neurons were used.

Whether sensory neurons were damaged by ischemia was analysed by a model with only the binary factor ischemia. *p*-values ≤ 0.05 were considered statistically significant. Graphs were generated using GraphPad Prism 9 (Graphpad Software Inc., San Diego, CA, USA) and arranged in CorelDraw 22 (Corel Corporation, Ottawa, ON, Canada).

## Figures and Tables

**Figure 1 ijms-24-02516-f001:**
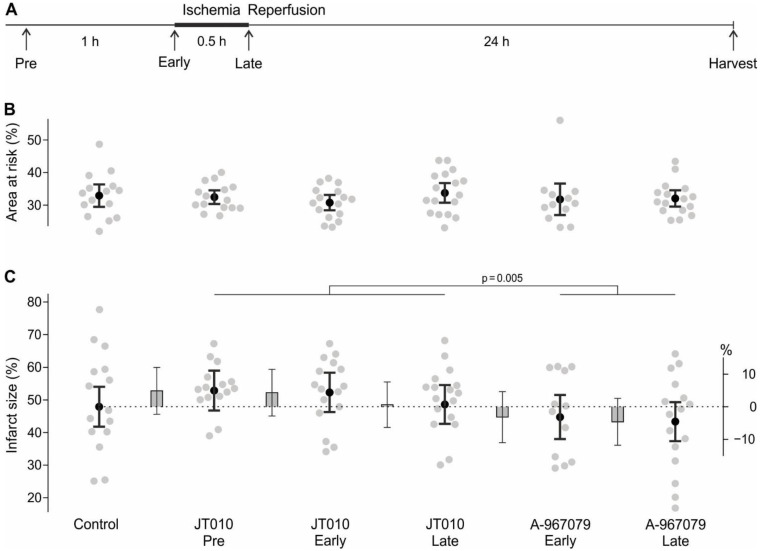
TRPA1 agonists and antagonists had small opposing effects on myocardial infarct size. (**A**) Myocardial IR experimental protocol in rats with time points of treatment indicated by arrows. Rats were treated either 60 min (Pre) or 5 min prior to ischemia (Early) or 5 min prior to reperfusion (Late) with the TRPA1 agonist JT010 (0.4 mg/kg) and Early or Late with the TRPA1 antagonist A-967079 (6.2 mg/kg) i.v. (**B**) Area at risk (% of left ventricle). Each grey symbol represents the result of a single rat, black dots with error bars represent means ± 95% CI. (**C**) Infarct size in percent of left ventricle area at risk 24 h after myocardial IR. As above, grey symbols represent results of single rats and black dots means ± 95% CI. Estimated differences to control are shown as grey bars ± 95% CI and refer to the right *y*-axis.

**Figure 2 ijms-24-02516-f002:**
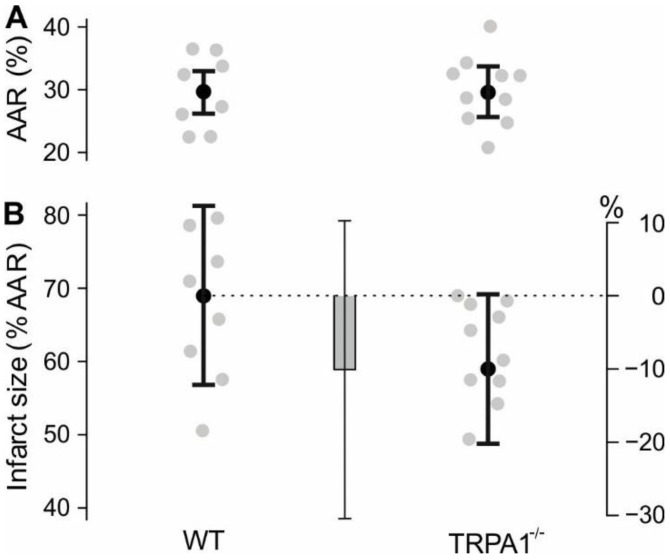
No evidence for a substantial effect of TRPA1 deletion on myocardial infarction. (**A**) Area at risk (% of left ventricle, AAR). Every grey dot represents one mouse, black dots with error bars are means ± 95% CI. (**B**) Infarct size in percent of left ventricle area at risk, determined after 24 h. Every grey dot represents one mouse, the estimated means ± 95% CI are shown as black dots. The estimated difference between the groups is shown as grey bar ± 95% CI.

**Figure 3 ijms-24-02516-f003:**
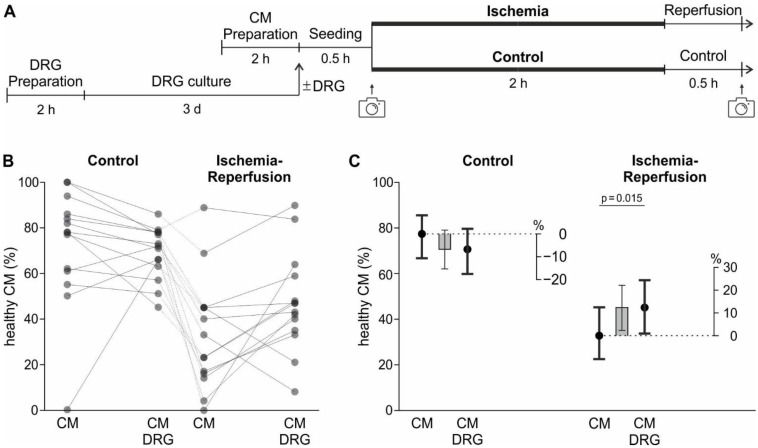
Sensory neurons increased the survival of cardiomyocytes in a model of IR. (**A**) Experimental timeline for the cardiomyocytes, including timing of image acquisition for survival analysis. In experimental groups with sensory neurons from dorsal root ganglia (DRG), these were prepared and seeded three days earlier. (**B**) Percentage of healthy CM at the end of the cellular model for IR, with 2 h of mimicked ischemia and 0.5 h mimicked reperfusion. Every dot represents the mean of one experiment, and those experiments conducted in parallel are connected. (**C**) Estimated survival probabilities for each group are shown as black dots ± 95% CI and estimated differences are shown as grey bars ± 95% CI.

**Figure 4 ijms-24-02516-f004:**
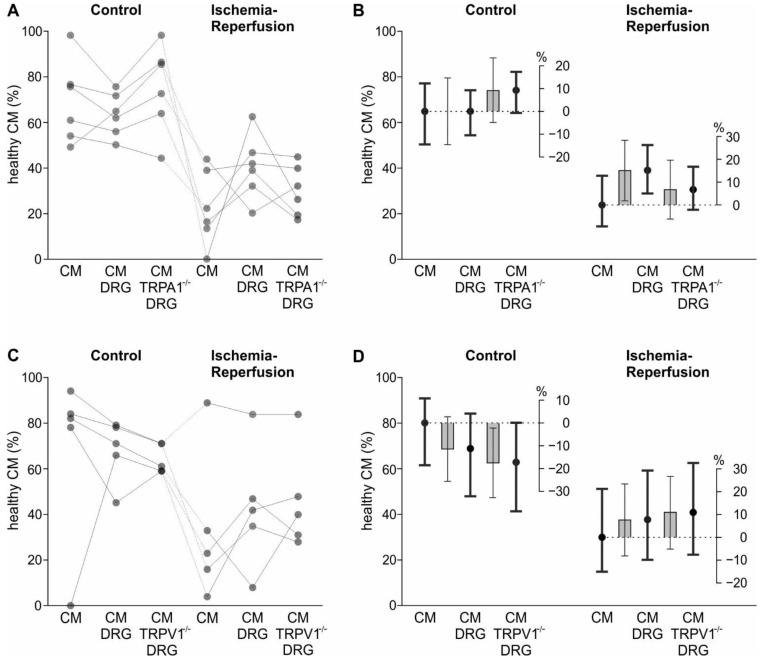
Involvement of sensory neuron TRP channels in increased CM ischemia-reperfusion tolerance. (**A**,**C**) Percentage of healthy CM at the end of the cellular model for IR, with 2 h of mimicked ischemia and 0.5 h mimicked reperfusion. Every dot represents the mean of one experiment, and those experiments conducted in parallel are connected. Experiments were conducted with WT sensory neurons from dorsal root ganglia (DRG) and with TRPA1^−/−^ sensory neurons (*n* = 6 experiments). The WT sensory neurons are contained in the 14 independent experiments of Figure 3A,B. (**B**) Estimated survival probabilities for each group are shown as black dots ± 95% CI, estimated differences are shown as grey bars ± 95% CI. (**C**) Independent experiments, conducted with WT or TRPV1^−/−^ sensory neurons. (*n* = 5 in panel C). Again, the WT sensory neurons are contained in the 14 independent experiments of Figure 3A,B. (**D**) Survival probabilities and estimated differences as in panel C.

## Data Availability

The data presented in this study are available on request from the corresponding author.

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
