# Peer review of "TRPA1 as Target in Myocardial Infarction"

_ijms, 2023, doi:10.3390/ijms24032516_

Round 1
Reviewer 1 Report
In the present study by Hoebart et al that is under consideration in International Journal of Molecular Sciences transient receptor potential cation channel subfamily A member 1 (TRPA1) as a potential pharmacological target in myocardial infarction is investigated. The authors performed in vivo animal experiments in which they used agonist and antagonist of TRPA1 as well as used TRPA1 knockout mice to examine the role of TRPA1 in myocardial infarction. No significant differences were found between treated or TRPA1-KO and control groups regarding the infarct size. However, TRPA1 agonist and antagonist treatments were observed to increase or decrease infarct size, respectively. The authors also performed in vitro experiments to investigate the effects of sensory neurons on cardiomyocyte survival in ischemia-reperfusion model. They found some improvement in the cardiomyocyte survival in the presence of TRPA1-expressing sensory neurons indicating a partial role of TRPA1 in ischemia-reperfusion injury. The authors concluded that TRPA1 may be neither suitable for targeting when treating myocardial infarction, nor contraindicated when using for other diseases.
The paper is well-written and provides interesting aspects on TRPA1 in ischemic injury of heart. The findings are important in particular as TRPA1 is considered to be a potential new pharmacological target. However, there are some questions that should be answered by additional data and extended discussion before further consideration.
Major/minor comments:
1. The authors provided clear evidence for the lack of protective effect of TRPA1 agonists or antagonists regarding the infarct size. However, no alternative end points were given in the manuscript. As TRPA1 may play a role in cardiac arrhythmias (PMID: 30351402, 27746315, 26718787), it would be impressive to provide some information about the ventricular arrhythmia (incidence, duration) that occurs during the experiments if possible. Are there are differences between the groups?
2. Along the same line, is there any data on laboratory biomarkers that are relevant at this time point (e.g. troponin)? That might be another alternative endpoint for detecting any effects of treatments on myocardial infarction.
3. The authors included data on exclusion and mortality of animal experiments. Treating animals with A-96 (TRPA1 antagonist) increased overall mortality by eye which might have an impact on a seemingly better outcome for A-96 treated animals. Despite the fact that the difference was not statistically significant regarding the exclusion, are there any explanations or literature data for the observed phenomenon? Please discuss it.
4. The authors described a manual method for determining cell viability. Please include details about the number of blinded individual examiners, the counted cell numbers, etc. In addition, cell viability assay based on the metabolic activity of cells should be considered to confirm the results.
5. Representative images of infarct size from each group would be an impressive addition to in vivo data.
Author Response
submitted as word file

Reviewer 2 Report
Hoebart and colleagues investigate the potential contribution of TRPA1, presumably located on sensory neurons, on cardiomyocyte death in the setting of ischemia-reperfusion and permanent coronary artery ligation. Using a combination of in vivo and cell culture models, together with pharmacologic and genetic strategies to modulate the receptor, the authors conclude that TRPA1 does not play a major role in determining cardiomyocyte fate.
The strengths of this manuscript include the rigor of the study design, together with the robust a priori plan for the statistical analysis. However, several important questions must be addressed.
Major Comments:
11. The authors state on lines 538-39 that there were “no substantial deviations from the assumptions” made in developing the statistical plan. This statement appears to be incorrect: the anticipated infarct size in untreated control rats was 65% of the risk region, with a 15% reduction in infarct size with treatment considered to be “relevant” (lines 530-31). However, the actual infarct size in control rats was 47.9%: i.e., 17.1% lower than anticipated, and within the range proposed to reflect a relevant and successful therapeutic intervention. Is there a reason for the substantial difference in anticipated versus actual infarct sizes in control rats? Perhaps the assumption was based on data from pentobarbital-anesthetized rats rather than animal anesthetized with ketamine + xylazine? What impact does this substantial deviation from the initial assumption have on the interpretation of the data?
22. Although differences in infarct size data between agonist-treated rats, antagonist-treated rats, and TRPA1-/- mice did not differ when compared with the corresponding control groups, there was weak evidence of a group effect in the rat model (p=0.071), and a 10% reduction in mean infarct size in TRPA1-/- mice versus wild-type. Accordingly, these data could be interpreted as evidence that, despite the prospective power analysis, the infarct size protocols were under-powered.
Minor Comments:
33. Please include original, representative images of TTC-stained heart slices for all groups in both the rat and mouse protocols.
44. Please provide information on the specificity of JT010 and A-967079 for TRPA1. Do the agonist and antagonist act on other ion channels? Other molecular targets?
55. Why was ischemia-reperfusion utilized in the rat model and in the cell culture models, while mice were subjected to permanent coronary artery ligation?
Author Response
Reviewer 2
Hoebart and colleagues investigate the potential contribution of TRPA1, presumably located on sensory neurons, on cardiomyocyte death in the setting of ischemia-reperfusion and permanent coronary artery ligation. Using a combination of in vivo and cell culture models, together with pharmacologic and genetic strategies to modulate the receptor, the authors conclude that TRPA1 does not play a major role in determining cardiomyocyte fate.
The strengths of this manuscript include the rigor of the study design, together with the robust a priori plan for the statistical analysis. However, several important questions must be addressed.
Major Comments:
- The authors state on lines 538-39 that there were "no substantial deviations from the assumptions" made in developing the statistical plan. This statement appears to be incorrect: the anticipated infarct size in untreated control rats was 65% of the risk region, with a 15% reduction in infarct size with treatment considered to be "relevant" (lines 530-31). However, the actual infarct size in control rats was 47.9%: i.e., 17.1% lower than anticipated, and within the range proposed to reflect a relevant and successful therapeutic intervention. Is there a reason for the substantial difference in anticipated versus actual infarct sizes in control rats? Perhaps the assumption was based on data from pentobarbital-anesthetized rats rather than animal anesthetized with ketamine + xylazine? What impact does this substantial deviation from the initial assumption have on the interpretation of the data?
We agree that the observed infarct size fraction of the area at risk was smaller than anticipated, and to a relevant degree. Both the chosen anaesthesia as well as the rat strain might contribute to this result. However, while the infarct size was lower than anticipated, the standard deviation was very close to what was expected: pooled over all groups 11.97. For statistical power the difference between the group means and the variances of each group are important, while the absolute mean values in each group are irrelevant. Therefore, the statistical power to detect 15% difference was approximately as expected despite lower absolute mean values. The a priori determined relevant effect size (15% reduction) represents a larger fraction of the 47.9% infarct size. This does not necessarily change the interpretation, but should indeed be noted. We added: “The observed infarct size, in the absence of any pharmacological intervention, was lower than anticipated when the study was planned, the pooled variances matched the expectation. Possible explanations might be the chosen anaesthetic protocol as well as the rat strain”.
- Although differences in infarct size data between agonist-treated rats, antagonist-treated rats, and TRPA1-/- mice did not differ when compared with the corresponding control groups, there was weak evidence of a group effect in the rat model (p=0.071), and a 10% reduction in mean infarct size in TRPA1-/- mice versus wild-type. Accordingly, these data could be interpreted as evidence that, despite the prospective power analysis, the infarct size protocols were under-powered.
We agree that there is a tendency towards reduced infarct size fraction for TRPA1 inhibition or absence. However, neither estimate nor the range of the confidence intervals thereof indicate a relevant change. So even in case a substantially larger sample size would result in statistical significance, the conclusion would not change much. TRPA1 would then, at best, have a biologically marginal effect in infarct size.
Minor Comments:
- Please include original, representative images of TTC-stained heart slices for all groups in both the rat and mouse protocols.
Representative images of the infarct size have been added as new Figure A1.
Figure A1: Representative images of staining for area at risk and infarct size. Myocardial infarction analysis by triphenyltetrazolium chloride staining. Representative images are provided for (A) rats and (B) mice.
- Please provide information on the specificity of JT010 and A-967079 for TRPA1. Do the agonist and antagonist act on other ion channels? Other molecular targets?
JT010 is the most potent known TRPA1 agonist with a reported EC50 of 0.65 nM, several log steps below other common agonists (Takaya et al., J. Am. Chem. Soc. 2015, 137:15859–15864). They investigated specificity and found no relevant activation of other targets activated by electrophiles, including TRPV1, TRPV3, TRPV4, TRPM2, TRPM8 and TRPC5. We retested this in cellular assays and were the first to probe this agonist in human subjects, which renders it with slightly less potency than originally reported still the best TRPA1 agonist (Heber et al., J Neurosci 2019; 39:3845-3855).
A-967079 is a potent and specific antagonist of rat TRPA1 with an IC50 about 101 nmol/L in electrophysiology and 289 nM in calcium imaging (Chen et al., Pain 2011; 152:1165-1172). They report that A-967079 is “>1000-fold selective over other TRP channels, and is >150-fold selective over 75 other ion channels, enzymes, and G-protein-coupled receptors”. The binding site on TRPA1 has been determined (Banzawa et al., J Biol Chem 2014; 289:31927-31939). We could demonstrate that the employed concentration quantitatively inhibits TRPA1 agonism, again in cellular assays but also in human subjects (estimated inhibition 96%, again Heber et al., J Neurosci 2019; 39:3845-3855).
The respective paragraph has been extended and the original publications on the two substances have been added to the manuscript.
- Why was ischemia-reperfusion utilized in the rat model and in the cell culture models, while mice were subjected to permanent coronary artery ligation?
The most commonly used model for MI in vivo is surgical ligation of the left anterior descending coronary artery (LAD) with two predominant approaches: permanent ligation, where the LAD is permanently occluded with a suture, or ischemia-reperfusion, where the LAD is temporarily occluded before removing the suture to restore blood flow. In general, ischemia-reperfusion salvages some of the area at risk. Thus, the scar is smaller and includes reperfusion injury, an additional smaller, second wave of necrotic damage.
In our study ‘late’ TRPA1 modulation, that is after ischemia but before reperfusion, did not alter infarct size. Therefore, we decided to use a permanent ligation model of MI in mice, which focuses on ischemic damage.

Reviewer 3 Report
In this manuscript, Hoebart et al. investigated whether activation, inhibition or absence of TRPA1 affects infarcts and to explore underlying mechanisms. Although authors showed a large amount of data to clarify their hypothesis, some results should be supplemented to further improve the quality of this work. Detailed comments are listed below,
1 the protein level of TRPA1 in gene knock out animal shall be tested and provided.
2 the infarct size staining for animals shall be added.
3 why was sensory neurons from dorsal root ganglia (DRG) of TRPA1-/- animals included in this study in figure 3?
4 the TRPA1 siRNA should be used as a control arm?
5 are there any better semi-qualitative methods to cardiomyocyte survival instead of morphology?
Round 2
Reviewer 1 Report
All issues have been appropriately addressed.
Reviewer 2 Report
All concerns raised in my previous critique have been adequately addressed.
Reviewer 3 Report
I am fine with this revision.